# Vitamin D Restores Skeletal Muscle Cell Remodeling and Myogenic Program: Potential Impact on Human Health

**DOI:** 10.3390/ijms22041760

**Published:** 2021-02-10

**Authors:** Clara Crescioli

**Affiliations:** Department of Movement, Human and Health Sciences, Section of Health Sciences, University of Rome “Foro Italico”, Piazza L. de Bosis 6, 00135 Rome, Italy; clara.crescioli@uniroma4.it; Tel./Fax: +39-06-36733395

**Keywords:** vitamin D, skeletal muscle cells, cell remodeling, health

## Abstract

Skeletal muscle cells, albeit classified as vitamin D receptor (VDR)-poor cells, are finely controlled by vitamin D through genomic and non-genomic mechanisms. Skeletal muscle constantly undergoes cell remodeling, a complex system under multilevel regulation, mainly orchestrated by the satellite niche in response to a variety of stimuli. Cell remodeling is not limited to satisfy reparative and hypertrophic needs, but, through myocyte transcriptome/proteome renewal, it warrants the adaptations necessary to maintain tissue integrity. While vitamin D insufficiency promotes cell maladaptation, restoring vitamin D levels can correct/enhance the myogenic program. Hence, vitamin D fortified foods or supplementation potentially represents the desired approach to limit or avoid muscle wasting and ameliorate health. Nevertheless, consensus on protocols for vitamin D measurement and supplementation is still lacking, due to the high variability of lab tests and of the levels required in different contexts (i.e., age, sex, heath status, lifestyle). This review aims to describe how vitamin D can orchestrate skeletal muscle cell remodeling and myogenic programming, after reviewing the main processes and cell populations involved in this important process, whose correct progress highly impacts on human health. Topics on vitamin D optimal levels, supplementation and blood determination, which are still under debate, will be addressed.

## 1. Introduction

The pleiotropic extra-skeletal effects of vitamin D are increasingly acknowledged. This compound, which is nutritionally classified as a fat-soluble vitamin, acts like a steroid hormone (via genomic and non-genomic mechanisms) and controls the function of many non-skeletal tissues and cells affecting human health and quality of life. Indeed, sufficient levels of vitamin D are essential to maintain whole-body homeostasis and health as optimally as possible, from fetal to old life [1], whereas vitamin D inadequacy is known to increase the prevalence of numerous diseases (i.e., diabetes, cancer, autoimmune and cardiovascular pathologies), including skeletal muscle diseases [2]. The widespread effect of vitamin D relies on the extensive presence of the vitamin D receptor (VDR), that is expressed virtually by every human tissue and nearly by all nucleated cells, although at variable concentrations [3,4,5]. Beyond kidneys, bones, and intestines, identified as the “classical” target tissues, malignant, immune, and smooth muscle cells are known to be “non-classical” targets, under vitamin D fine-tuned control. Noticeably, growing evidence supports critical effects of vitamin D also onto so-called VDR-poor cells, such as skeletal muscle cells. The biology and function of striated cells, despite low VDR expression, are exquisitely regulated by vitamin D, within either physiologic or pathologic contexts [6,7,8,9,10]. A wide spectrum of findings highlights the link between vitamin D deficiency and increase of skeletal muscle cell wasting, which turns into loss of tissue integrity/function, and, finally, ends in disease development [6,11,12]. Importantly, proper skeletal muscle cell remodeling is fully recognized as a key process to warrant tissue adaptation, recovery, and homeostasis [13,14], highly impacting general health status. Skeletal muscle cells are, indeed, considered actively determinant to drive biomolecular and intracellular processes toward a fully functional remodeling, in response to microenvironmental changes and demand. In this scenario, vitamin D supplementation could represent an optimal approach to maintain or restore skeletal muscle cell remodeling and tissue integrity, but data from clinical trials in humans are still inconclusive [15]. This review aims to overview how vitamin D can orchestrate skeletal muscle cell remodeling and myogenic program, after recalling the main processes and cell populations involved in this important process, whose function is not limited to meet hypertrophic needs. Given the importance of adequate vitamin D levels to maintain healthy conditions, topics on vitamin D optimal level, blood determination and supplementation will be addressed, underlining the debate still present on these issues.

## 2. Introducing Vitamin D: A Nutrient, a Hormone and a Rapid Regulatory Factor

Diet and sun exposure are the main sources of this vitamin, which, historically, was classified as “D” because it was the fourth discovered in the vitamin sequence [16]. The two main forms vitamin D3 (cholecalciferol) and vitamin D2 (ergocalciferol), from animal and vegetable origin, respectively, share similar metabolism and features. Upon sun exposure, 7-dehydrocholesterol, present in the skin, is converted to vitamin D3; vitamin D from dietary sources is transported in chylomicrons to the bloodstream. Circulating vitamin D and D-metabolites are mainly bound to vitamin D-binding protein (DBP) and, to a lesser extent, to lipoprotein and albumin, with only less than 1% circulating in free form [17]. The first enzymatic transformation in the liver by D-25 hydroxylase (CYP24A1) produces inactive 25-hydroxyvitamin D3, or 25(OH)D; the second enzymatic transformation in the kidneys by D-1 hydroxylase (CYP27B1) converts 25(OH)D to 1,25(OH_2_)D (or calcitriol), the biologically active form [18]. The past studies on the structure/function clarified that the affinity of 1,25(OH_2_)D for VDR is about 500 times more than 25(OH)D, albeit the circulating level of the inactive form is about 1000 times higher and more stable [19], likely representing a natural reservoir. The latter issue is relevant when dealing with vitamin D level determination, as addressed later in this review. Nowadays, the important pleiotropic extra-skeletal effects of vitamin D are well established in relation to their broad effects mediated by VDR, which is virtually ubiquitously expressed and upregulated by the ligand, through intronic and upstream enhancers [20,21]. Vitamin D signaling is mediated by classical genomic mechanisms through VDR heterodimerization with 9-*cis*-retinoic acid receptor (RXR) to form a dimeric complex VDR:RXR, which directly targets gene promoter regions, the vitamin D response elements (VDREs), to up- or down-regulate expression of a multitude of genes [22]. In addition, non-genomic vitamin D mechanisms, eliciting VDR translocation in plasma membrane through plasmalemma microdomains (or caveolae) highly specialized for macromolecule transcytosis, are known to rapidly activate transmembrane signal transduction pathway/intracellular cascades (within seconds to minutes) [23]. A membrane-associated receptor mediating rapid, non-genomic effects has been also described; nevertheless, this issue is still disputed [24]. Figure 1 summarizes vitamin D metabolism and genomic, non-genomic signaling. Thus, “the nutrient, the hormone and the rapid regulating factor” vitamin D can finely impact a broad spectrum of biological activities and, consequently, human health. Out of all these functions, vitamin D actions onto skeletal muscle cell remodeling will be highlighted, as this molecule can control almost each stage involved in this process, which remarkably goes beyond muscle mass repair and size.

## 3. Skeletal Muscle Cell Remodeling: Not Only a Matter of Size

Skeletal muscle shows a good level of plasticity and undergoes constant remodeling in response to a variety of environmental, physiological or pathological stimuli, i.e., nutrition, exercise and poor health status. Cell remodeling is a critical and complex process under multilevel regulation, orchestrated mainly by the satellite niche, and by non-myogenic cells, protein synthesis/breakdown, gene transcriptional control, as recently summarized [14]. The continuous turnover of cell population aims to remove old/damaged cellular components and substitute them with new ones, allowing a constant tissue renewal and regeneration [13]. This process, which involves the activation of stem cell population and the increase in protein synthesis rate—i.e., after protein ingestion or recovering from resistance exercise—is often seen only through the lens of myofiber repair and hypertrophy.

The integrated remodeling processes elicit not only size modification, which, indeed, can be reached independently of satellite cell response [25,26], but also fiber-type and metabolic adaptation, to maintain tissue function, physical performance, and health. 

So far, beyond mass hypertrophy or myofiber repair after injury, the concept of cell remodeling to maintain tissue function and health for non-hypertrophic functions has taken place. Satellite cells, by replacing and compensating old components, i.e., following stimuli like nutrition or exercise, would continuously refresh myocyte transcriptome and correct genetic information to the intracellular machinery dedicated to protein synthesis. Thus, a renewed cell proteome has likely provided for the optimal maintenance of tissue function and integrity [13]. Figure 2 outlines these processes.

Conversely, in the presence of gene expression deregulation, following damaged DNA accumulation, i.e., due to aging or sedentary lifestyle, the rate of tissue remodeling decreases and allows tissue misfunction, with detrimental effects on health [27]. An intriguing hypothesis is that hypertrophic and non-hypertrophic muscle remodeling collaborate and act in different temporal windows—short-term (hours) and long-term (days)—despite the same time of exposition to nutrient or exercise stimuli [13]. Different types of cells residing within muscle such as vascular cells, myoendothelial cells, fibroblasts, pericytes and progenitor populations—i.e., interstitial and side population cells—seem to participate in the myogenic program [28]. The latter, if properly working, ends in a balanced compensatory regeneration; if not, it allows maladaptive processes, leading to muscle fibrosis, fat infiltration and disease development. Particular attention is given to the interplay of satellite cells with fibro/adipogenic progenitors (FAP), a mesenchymal cell population resident in the interstitium, that originates adipocytes and fibroblasts, regulates components of extracellular matrix (collagens, fibronectin, laminin) and controls satellite cell differentiation [29,30,31]. The major evidence on the cross-talk between satellite cells and FAP essentially derives from animal studies, which not always resemble the processes occurring in humans (due to specie-specific differences in time frames and cellular components) [28]. However, satellite cell–FAP functional interactions have been reported in humans by studies on prolonged resistance exercise training effects or on myopathy [32,33,34,35]. Upon aberrant regulation, FAP, whose function is normally dedicated to muscle remodeling and regeneration, promote inflammation and tissue fibrosis [36]. The role of FAP likely depends on their phenotypes, which can be pro-regenerative and pro-apoptotic or pro-fibrotic and anti-apoptotic. The first one is associated with cell senescence/apoptosis/clearance in the remodeling/regenerating process, as occurs in exercise-induced skeletal muscle remodeling; whereas the pro-fibrotic/anti-apoptotic phenotype is linked to intramuscular fibrosis, impaired regeneration, tissue stiffness and contractile force reduction [36,37,38]. This phenomenon emerges quite promisingly to be potentially translated in clinics, i.e., intervening with some substances to restore FAP senescence in myopathy significantly ameliorates the therapeutic effect of exercise [36]. Skeletal muscle biopsies from subjects with type 2 diabetes (T2D) have been recently described to retain a higher level of FAP content with significant changes in FAP population, due to an increase in the pathogenic phenotype FAP^CD90+^, which is the cellular driver to muscle niche degeneration found in diabetes [39]. Within this multifaceted system, the balance between some positive myogenic regulatory factors (i.e., myogenin, calcineurin, insulin-like growth factor (IGF), desmin, myogenic factor 5 (Myf5), muscle-specific regulatory factor 4 (Mrf4) and myoblast determination protein (MyoD)) and biomediators of cell/tissue wasting (i.e., myostatin, tumor necrosis factor (TNF) α and ubiquitin pathway components) is critical for myoblast determination from progenitor satellite cells, and, in sequence, for correct myotube maturation and fusion, as extensively described elsewhere [6,13,28]. Whenever this system is compromised or damaged, i.e., due to satellite population loss or to intracellular signal missing, skeletal muscle remodeling fails and leads to pathological conditions, i.e., myopathy or degenerating diseases. Thus, interventions to maintain efficient muscle remodeling/regenerative programs are recommended for prevention or therapeutic approaches.

## 4. Vitamin D Impacts the Myogenic Program and Cell Remodeling toward Restored Functions

The role of vitamin D in the development, maintenance and regeneration of musculoskeletal system integrity and function is widely recognized. Contractility, strength and postural stability are known to associate with circulating vitamin D levels; generally, skeletal muscle weakness is seen as the common symptom of clinical vitamin D deficiency [40,41,42,43]. Indeed, the ability of vitamin D to impact muscle fiber morphology composition and muscular structure has been known since quite long ago [44,45]. As from pioneering studies onto human muscle biopsies, vitamin D deficiency associates with inter-fibrillar space enlargement, fat infiltration, fibrosis, glycogen granules and atrophy of fast-twitch type II muscle fibers [46,47,48,49]. Significant effects of vitamin D ability to correct and reverse many muscular defects, by interacting with its receptor, have been reported, albeit some controversies have existed on VDR presence in skeletal muscle [3]. To date, the nuclei of skeletal muscle tissue and isolated human myoblasts and myotubes express functional VDR, even if the expression level varies upon maturation stage, age and detection methodology [50,51,52,53]. Following exercise-induced muscle damage, VDR expression is reported to increase and interfere with pro-inflammatory cytokine gene expression, simultaneously altering several intracellular signaling responses to stress stimuli toward repair process, i.e., 5′ adenosine monophosphate-activated protein kinase (AMPK), mitogen-activated protein kinases p38 and extracellular signal-regulated kinase (ERK) 1/2 [54]. Gene-targeting in vivo studies on VDR-null mutant mice documented alterations of myogenic differentiation factors, muscle cell differentiation pathways and abnormal fiber development/maturation (i.e., smaller fiber diameters) [55,56]. Accordingly, in vitro investigations showed that myoblasts bearing siRNA-silenced VDR expression do not undergo myotube differentiation [57]. Thus, several data in animal and human cells pointed toward a positive role vitamin D/VDR signaling has in skeletal muscle remodeling, by interfering with the different factors and processes involved. VDR is expressed and upregulated upon ligand binding in satellite cells, which are the main population responsible for muscle turnover compensation and remodeling in adult life, as previously addressed [58]. Noticeably, vitamin D/VDR signaling impacts the myogenic program throughout all stages, from cell commitment, increasing Myf5 and MyoD, the “gatekeepers” to enter the myogenic lineage and determine cell identity, to myocyte fusion and myotube formation, enhancing the expression of myogenin (MyoG), and transcription factor MYC type II and MRF4, necessary to develop adult skeletal muscle phenotype [6,59,60,61]. Other pro-myogenic factors and processes are upregulated in satellite cells exposed to vitamin D, such as muscle troponin, which, beyond the canonical function in striated muscle cell contraction, plays novel important roles [58,60,62], myosin heavy chain I (MYH1), engaged in contraction-relaxation and in slow-to-fast/fast-to-slow fiber transition, or post-natal mitochondrial biogenesis [60,63]. Of note, in murine muscle cells, vitamin D enhances the intracellular pathways related to insulin-like growth factor (IGF) and fibroblast growth factor (FGF) signaling (the latter is undetectable in terminally differentiated cells) [58,60,64], pushing the myogenic program toward muscle regeneration. During the myogenic process, IGF-related signal suppresses myostatin (MSTN), the main negative regulator of muscle mass belonging to the TGF-β superfamily [65,66]. Thus, vitamin D, in addition to a direct suppression, indirectly inhibits MSTN, via IGF-dependent signal and follistatin, a MSTN inhibitor whose signaling is involved in the regulation of satellite cell myogenic potential [58,60,67]. In human cultured myoblasts, vitamin D significantly alters the oxygen consumption rate, affecting mitochondria volume and branching [68]. Interestingly, VDR has been reported to translocate into skeletal muscle cell mitochondria and impact bioenergetics [69], promoting the mediators of mitochondrial biogenesis and fusion—such as MYC, MAPK13, optic atrophy protein 1 (OPA1) and endothelial PAS domain-containing protein 1 mRNA—and decreasing mitochondrial fission mediators—such as mitochondrial fission 1 protein (Fis1) and dynamin-1-like protein (Drp1) [68]. Nowadays, mitochondria are considered the cellular sensors of stress and energy demand, finely regulated by a complex network of integrated signaling; a balanced mitochondrial activity is undeniably essential to mediate a healthy adaptive genomic reprogramming during skeletal muscle cell remodeling [70]. Noticeably, the dynamic remodeling of mitochondria in response to exercise retains the potentiality as a therapeutic approach in myopathy, dystrophy, T2D, chronic muscle disuse and age-related sarcopenia [71]. Some of the main vitamin D effects onto cell remodeling and myogenic programming are summarized in Table 1.

In isolated animal cells, the maximum oxygen consumption rate and ATP generation coupled to respiration increase upon vitamin D incubation; in line with in vitro evidence, in vitamin D deficient humans, the maximal mitochondrial oxidative phosphorylation rate rises upon cholecalciferol treatment [68,72,73,74]. It is known, since quite long ago, that animal muscle cells exposed to low vitamin D show increased reactive oxygen species (ROS), cytotoxicity and failure in Ca^2+^ transport [75]. Moreover, a prolonged status of hypovitaminosis D is likely to ablate VDR expression with a drastic reduction or even absence of cell remodeling associated with upregulation of muscle atrophy markers, such as muscle RING-finger protein-1 (Murf1) and muscle atrophy F-box/atrogin-1 (MaFbx) [74,76,77]. Genomic and non-genomic mechanisms have been proposed to explain the effect of vitamin D onto mitochondria, involving the proteins for electron transport and the enzymes dedicated to the tricarboxylic acid cycle [78], even if conclusive studies in skeletal muscle cells are still missing. Independently of the involved paths, altogether these observations strongly support that the mitochondrial activity and cell remodeling could be restored by a vitamin D-enriched diet or supplementation. This might raise particular interest, considering first that mitochondrial respiratory chain dysfunctions and ROS generation are critical steps in human disease development associated with muscle atrophy, and second, not less important, that mitochondria dynamic remodeling potentially represent a therapeutic opportunity, as previously commented in this review [71]. The experimental condition mimicking vitamin D deficiency upregulates adipogenic factors like peroxisome proliferator-activated receptor (PPAR)γ2 and fatty acid binding protein 4 (FABP4), both known to promote transition to adipogenesis. The transition of satellite cells to the adipocytic phenotype, the increased synthesis/level of triacylglycerol within skeletal muscle and the consequent intramyocellular depot of triglycerides are looked at as the main triggers of glucose intolerance and increased risk of metabolic disease [79,80,81,82]. Of note, the process toward adipogenesis can be reverted by adding adequate vitamin D concentration to skeletal muscle cells. To date, vitamin D induced opposite effects on cell differentiation fate, stating decreased expression of myogenic regulatory factors; these are also present in literature [50]. Indeed, some authors reported a robust inhibition by vitamin D onto human myoblast proliferation/differentiation and myotube formation along with a significant decrease in some myogenic regulatory factors, including MyoD and MyoG, and no changes in MSTN [50]. This effect occurred in association with induction of signaling paths promoting myoblasts quiescence (FOXO3 and Notch) and self-renewal of activated satellite cells. It is conceivable that some discrepancy in the results is species related, as most of the studies are performed in animal cells. 

Nevertheless, despite some controversy, the importance of vitamin D-induced control onto the skeletal muscle staminal niche is acknowledged. So far, the need to continue investigations is fully highlighted as a key point to further understand skeletal muscle biology and, consequently, make interventional decisions to maintain or restore tissue homeostasis and health [10]; i.e., it is conceivable that hypovitaminosis D promoting satellite maladaptive remodeling affects, in turn, their interplay with other critical cell populations, such as FAP, within this complex network. From these observations, the hypothesis that sufficient vitamin D intake, from fortified foods or supplementation, to support or correct skeletal muscle cell remodeling and skeletal muscle function has taken place. However, vitamin D measurement and supplementation are still controversial issues, as addressed in the following paragraph.

## 5. Vitamin D Supplementation: Where Are We?

Vitamin D insufficiency is a worldwide phenomenon. Albeit many factors, including genetic and environmental, concur to vitamin D deficiency. Its development is mainly due to prolonged lower dietary intake, such as ovo-vegetarian or vegan diet, limited sunlight exposure, i.e., caused by shifts to indoor lifestyle, and inadequate renal conversion or intestinal absorption. The best time-range for sunlight exposure to get sufficient ultraviolet (UV) radiation for vitamin synthesis seems to be 5–30 min between 10 a.m. and 4 p.m., but it is very difficult to provide guidelines, considering the high variability of individual responsiveness and the potential risk of UV-induced skin cancer [83,84]. The recommended protective sunscreens have sun protection factors (SPF) from 15 on, but an SPF equal to or higher than 8 is enough to inhibit UV-dependent vitamin D synthesis [84]. Vitamin D3 (animal) or D2 (vegetable) from dietary sources show no substantial difference in their metabolic steps or gut absorption; they both increase the serum level of 25(OH)D—the stable metabolite used for blood quantification—even if some evidence indicates higher/longer-lasting 25(OH)D from D3 [85,86,87]. Vitamin D is naturally present in few aliments; thus, fortified foods, such as animal or plant-derived milks (from soy, almonds, oats), yogurt, breakfast cereals, orange juice and bread have been introduced to provide adequate levels [88]. Given that low vitamin D status is largely diffuse and, as discussed above, greatly impacts on general health, the need for supplementation has been taken in full consideration [89,90]. Supplementation, in fact, potentially represents the necessary intervention to reduce the risk of low vitamin D-induced pathologic status, including skeletal muscle diseases. Remarkably, it must be recalled that the good health of skeletal muscle promotes and supports a good general health status [11,12]. The required doses for supplementation likely depend on the need to respond to skeletal/extra-skeletal needs. A clear definition of insufficiency and deficiency is still a matter of debate, due to the several variables known to affect vitamin status—i.e., age, sex, general health status, sedentary habits/physical activity, ethnicity, just to mention some [91,92]. According to most guidelines, the serum 25(OH)D concentration should be higher than 50 nmol/L (50–100 nmol/L); a range 75 to 125 nmol/L is recommended by the Central European guideline [93,94,95]. While the need of preventing hormone deficiency is undeniable, clear indications and statements on vitamin D supplementation still need to be defined. Likely, the uncertainty regarding the supplementation mirrors the unsolved issues about vitamin D determination. As previously addressed, 25(OH)D measurement is the best indicator for clinical and diagnostic purposes in the general assessment of population or individual vitamin D status, as this metabolite is stable and comprises the total amount deriving from diet (D3 or D2) or dermal sources [94]. Concerns on quality assurance arise from the lack of well-defined standardization of assay methodologies and from intra- and inter-laboratory high variability [94,96,97].

Some standardizing protocols of 25(OH)D measurement have been proposed by the NIH-led international Vitamin D Standardization Program (VDSP), and other programs have been developed (i.e., National Institute of Standards and Technology or NIST) to decrease method variability and provide international standards, as exhaustively reported elsewhere [94,96]. Thus far, even after many efforts and the achievement of some results, the journey to reach fully accepted, reliable assessment in vitamin D measurement and supplementation is still long. In turn, the bias encountered in vitamin D determination/supplementation might explain, at least in part, the lack of consistent results from different trials as exhaustively reviewed [15]. Concerning vitamin D supplementation and skeletal muscle function, the preferred populations to investigate in trials would be the elderly or athletes, to ameliorate frailty and sarcopenia-related maladjustments or physical performance and recovery, respectively, but the results are still inconclusive; albeit some beneficial effects in muscle strength are reported in subjects with low basal D levels [98,99]. Given the high variability and related bias, the general conclusions in literature converge into the urgent need for well-designed research in larger sample sizes with adequate study length and study follow-up. 

## 6. Conclusions

Low vitamin D status is unquestionably associated with skeletal muscle cell maladaptation and tissue deterioration, with important consequences on general health status, given the role of skeletal muscle integrity onto whole-body homeostasis. A correct skeletal muscle cell remodeling in response to different stimuli within physiological or pathological contexts—i.e., after exercise, during aging or diseases—not only meets hypertrophic and reparative needs, but continuously refreshes cellular transcriptome and proteome towards optimal adaptive processes necessary for a constant muscle renewal. Considering that vitamin D intake can correctly promote myogenic programing and remodeling, beyond other beneficial effects, it could be the desired approach to preserve skeletal muscle integrity and function as optimally as possible. To date, the literature to support vitamin D supplementation is unsatisfactory, due to the inconclusiveness of data from studies or meta-analysis presenting bias, as is the use of different unstandardized measures. Hence, although general vitamin D effects on muscle have been known for quite a long time, paradoxically, the research on this topic is still in its infancy. Well-designed trials in restricted, well-defined groups are mandatory to limit bias and variability. Further characterization of the role of vitamin D/VDR signaling in skeletal muscle cell remodeling is necessary to reveal potential intracellular targets and limit muscle wasting/weakness. We could speculate that reliable in vivo and in vitro data, once well-defined and validated, would converge to possible translation in clinical application. In conclusion, a nutraceutical approach, through vitamin D supplementation, retains the potentiality to be an optimal, inexpensive and safe strategy for preventing/limiting/reversing muscle wasting and ameliorate human health. 

## Figures and Tables

**Figure 1 ijms-22-01760-f001:**
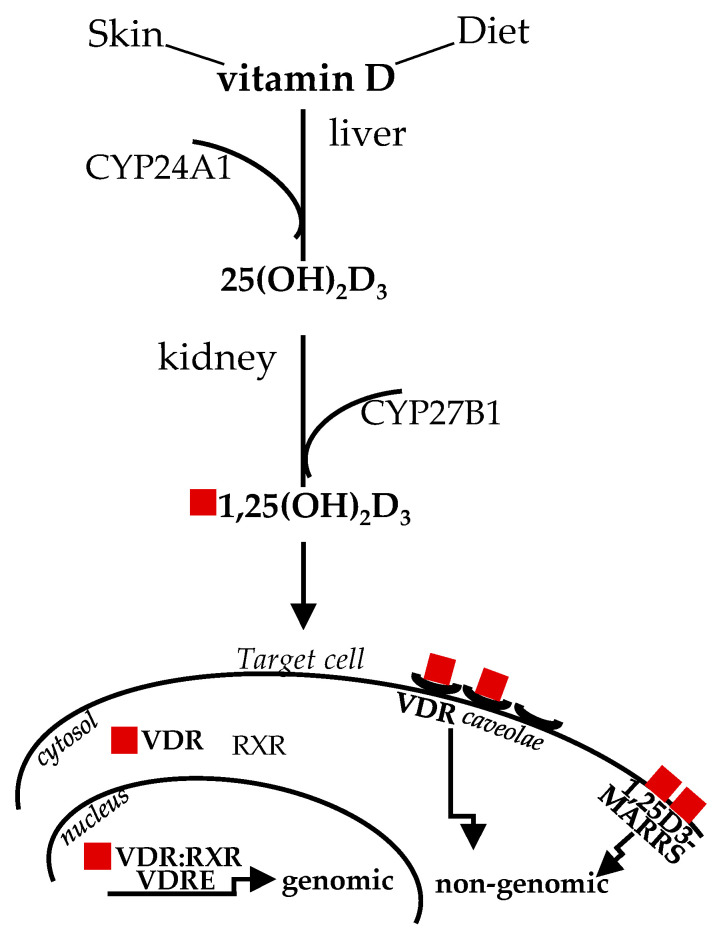
Vitamin D sources, metabolism and signaling. The active form of vitamin D derived by sun exposure or diet, after two enzymatic steps in the liver and kidney, binds to VDR in the cytosol to mediate genomic effects; alternatively, ligand binding, to caveolae-mediated or membrane-associated VDR, mediates rapid genomic responses. CYP24A1, D-25 hydroxylase; CYP27B1, D-1 hydroxylase; VDR, vitamin D receptor; RXR, with 9-cis-retinoic acid receptor; 1,25D3-MARRS, membrane-associated, rapid response steroid binding.

**Figure 2 ijms-22-01760-f002:**
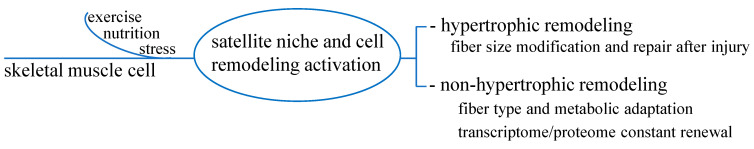
Hypertrophic and non-hypertrophic functions of skeletal muscle cell remodeling. In addition to hypertrophic function to satisfy the need of fiber size modification and repair, non-hypertrophic function seems to constantly refresh transcriptome/proteome renewal to warrant correct adaptative processes in response to stimuli such as exercise, nutrition or stress.

**Table 1 ijms-22-01760-t001:** Vitamin D enhances cell remodeling and myogenic program. Adequate levels of vitamin D support skeletal muscle cell remodeling and myogenic programming, pushing towards regeneration. Myf5, myogenic factor 5; MyoD, myoblast determination protein; MyoG, myogenin; MYCII, transcription factor MYC type II; Mfr4, muscle-specific regulatory factor 4; MYH1, myosin heavy chain I; IGF, insulin-like growth factor; FGF, fibroblast growth factor; MSTN, myostatin.

Myogenic Regulation	Function	Vitamin D	References
Myf5, MyoD	gatekeepers to enter myogenic lineage	+	[6,58,60,69]
MyoG, MYCII, Mfr4	adult cell phenotype development	+	[6,58,60]
Troponin, MYH1	promyogenic factors	+	[58,60]
IGF/FGF program	muscle cell regeneration	+	[58,60,69]
MSTN	myogenesis negative regulator	–	[6,58,60]
Follistatin	MSTN inhibitor	+	[6,58,60]
Mitochondria remodeling	biogenesis and fusion	+	[60,68]

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
