# Peer review of "Vitamin D Restores Skeletal Muscle Cell Remodeling and Myogenic Program: Potential Impact on Human Health"

_ijms, 2021, doi:10.3390/ijms22041760_

Round 1
Reviewer 1 Report
This is a timely and nicely written review from Dr. Crescioli, summarizing existing work on vitamin D on the skeletal muscle remodeling and regeneration. I only have a few minor comments/suggestions.
- In the graphical abstract and page 4, line 124, when the author describes the cellular constituents of the skeletal muscle, "vascular cells" and "myoendothelial cells" are used. Please clarify if by "vascular cells" you mean "endothelial cells"? Also, my understanding is that "myoendothelial cells", first reported by Tamaki et al. (PMID: 11994315), represent a rare population of myogenic cell type that shares gene signature of endothelial lineage. Thus, it should be classified as one of the "progenitor populations", like other rare myogenic cell lineages such as PW1+/Pax7- interstitial cells and Twist2+ cells.
- Minor suggestion: please consider adding a cartoon illustration summarizing the vitamin D metabolism and downstream signaling pathway at the second section of the manuscript.
- Another minor suggestion: please consider transforming current Figure 2 to a table and list the reference on the side of the downstream effectors.
- Please revise this sentence on line 93: "... and undergoes a constant remodeling...." (please remove "a")
- Please revise this sentence on line 126: "...seem to participate to the myogenic program." (please change "participate to" to "participate in")
Author Response
We thank the R1, whose suggestions and comments ameliorate the paper.
Please find as follows point-by-point responses.
- In the graphical abstract “vascular cell” is substituted with “vascular smooth muscle cell” and “myoendothelial cell” is changed to “endothelial cell”, as myoendothelial cells belong to progenitor population, as rightly observed by the Reviewer.
- A cartoon summarizing vitamin D metabolism and signaling paths is following the second section of the manuscript, as new figure 1 of the revised text; figure 1 legend comments the figure. The number of the following figure has been changed accordingly.
- Previous figure 2 is transformed in new table 1 in the revised manuscript.
- The sentence on previous line 93 has been corrected (“a” removed).
- The sentence on previous line 126 has been corrected ("participate to" is changed to "participate in").
Reviewer 2 Report
The authors described review the current data on IRE1 activation based on structure, XBP1 splicing and RIDD. Moreover, the authors discussed the functional diversity of IRE1. This paper is well written and organized.
I have only one remark.
Authors described “To date, vitamin D-induced opposite effects on cell differentiation fate, stating decreased expression of myogenic regulatory factors, are also present in literature (line 254-255). In conclusion part, authors described “To date, the literature to support vitamin D supplementation is unsatisfactory, due to the inconclusiveness of data from studies or meta-analysis presenting bias, as the use of different unstandardized measures”.
I think that it’s better to describe about reports on the opposite effects of vitamin D. I think this well deepen reader’s understanding. Therefore, I suggest that authors give a few details of research paper, for example ref#51.
Author Response
We are really grateful to R2 for the comments.
According to the suggestion, a rapid description of vitamin D opposite effects obtained in human and animal cells, referred to ref#51 is present and commented in the revised text, paragraph 4, lines 269-275.
Round 2
Reviewer 2 Report
Thank you for your effort.